# The Influence of the Mechanism of Double-Sided FSW on Microstructure and Mechanical Performance of AZ31 Alloy

Suna Cha [1], Hongliang Hou [1,2,3,*] and Yanling Zhang [1,2,3]

1   AVIC Manufacturing Technology Institute, Beijing 100024, China; suna_cha@163.com (S.C.); sam.hanlixi@gmail.com (Y.Z.)
2   Aeronautical Key Laboratory for Plastic Forming Technology, AVIC Manufacturing Technology Institute, Beijing 100024, China
3   Beijing Key Laboratory of Digital Plasticity Forming Technology and Equipment, AVIC Manufacturing Technology Institute, Beijing 100024, China
*   Correspondence: hou_hl@163.com; Tel.: +86-010-8570-1265

**Abstract:** In the friction stir welding (FSW) process, the final performance of weld joints is determined by microstructures influenced mainly by the heat input and mechanical deformation. In this research, the effects of FSW parameters, rotation speeds, and welding passes, on microstructure and mechanical properties of AZ31 alloy were systematically and comparatively studied. It was found that the microstructure at the joint center with multi-pass FSW could obtain a smaller average grain size compared with the single pass. The differences of the grain size were reduced significantly when the samples experienced the double-side FSW process. The mechanical performance results showed that the optimum strength (315 MPa) was achieved through the double-side FSW process with a rotation speed of 500 r/min and welding speed of 60 mm/min. The mechanism of the parameters and double-sided process on mechanical properties of the joint samples was elaborated.

**Keywords:** AZ31 alloy; friction stir welding (FSW); mechanical properties; recrystallization

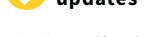



## 1. Introduction

The magnesium alloys have been widely used in the manufacture of aviation, automobile, and other fields, due to superior properties, such as high specific strength, good damping capacity, and easy recycle [1–5]. However, magnesium alloys have poor workability caused by the limited number of slip systems associated with a hexagonal, close-packed crystal structure [6]. Therefore, the majority of magnesium alloy products are fabricated through the casting technology because of the poor deformability [7]. Welding technology is essential for the development of manufactured products. Currently, fusion welding and hybrid welding are common methods used to join magnesium alloys [8]. However, the defects, e.g., hot cracking, residual stress, partial melting zone, and porosity generated by conventional fusion welding process [9], significantly deteriorate joints properties, which hinders the wide range of applications of magnesium alloys. As an advanced welding technology, friction stir welding (FSW) is conducted without fusion, solidification, and oxidation due to the characters of solid phase connection [10]. The limitation of joining magnesium alloys caused by the process of metallurgy and solidification in conventional welding technology can, therefore, be solved through the FSW process [11]. In the past decades, many scholars have carried out a lot of research focused on the effect of FSW tool pin geometry, process parameters on the joint microstructure mechanical, and the properties of aluminum alloys [12–16].

In FSW, the rotating tool is made to plunge and move transversely along the gap between the two workpieces. The requisite heat for the joint development is produced by the spin friction between the rotating tool and workpiece. The material abutting the joint line is soft, and the joint is produced in a solid state. The character of the solid-state process

enables FSW to weld dissimilar and difficult-to-weld materials [17–19]. Microstructures of FSW joints are composed of a weld nugget zone (NZ), thermo-mechanical affected zone (TMAZ), heat affect zone (HAZ), and base materials (BM), which are mainly dependent on parameters such as friction stir tool parameters, tool rotation speed, traverse speed (feed or welding speed), and tilt angle [20]. Plenty of researches have tried to grasp the system impact parameters on the mechanical properties and formation of microstructures in FSW joints. FSP can refine grain structures in the matrix of AZ31 magnesium alloy. Meanwhile, defects induced by the process of metallurgy and solidification in the conventional welding process can be eliminated [21]. Barmouz found that FSW can refine the second particle's size and enhance the dispersion distribution of the second particles for Cu/SiC composites [22].

Mohamed et al. [23] examined the effect of input parameters on tensile strength and hardness in FSW of AA6061 alloy and AZ31 alloy. Sankar et al. researched the influence of various operating parameters, namely tool rotation speed, feed, and tool diameter, on the mechanical properties of the FSW joint on AA6061 alloy [24–26]. Bai et al. verified that joint strength could be improved significantly by the application of ultrasonic vibration during the FSW [27]. Jayaprakash et al. found that the triangular tool offered better tensile strength and microhardness of the joint of AA5083 and AA7068 compared with the cylindrical taper tool of the investigation [28]. Up to now, some researchers have concluded several mechanisms for dynamic recrystallization (DRX) in FSW of aluminum alloys, which are included in discontinuous dynamic recrystallization (DDRX), continuous dynamic recrystallization (CDRX), and geometric dynamic recrystallization (GDRX) [29–34]. Additionally, it has been reported that single-pass FSW requires a careful selection of welding parameters as improper welding parameters give rise to the formation of defects (such as cavity, tunnel, and kissing bond defects) [35,36]. Kavitha et al. studied the effect of FSW parameters on the joint strength of AA7079 and AA8050 through a statistical technique of RSM, and found the preferred process parameters [37]. Compared to the single-pass FSW, the application of a second overlapping pass prolonged the DRX time and the DRX became sufficient, resulting in further grain refinement [38,39]. Chen et al. found that reversing the welding direction of the second overlapping pass enhanced the vertical flow, increasing the FSW strain in the NZ [40].

As discussed above, the effects and operational mechanism of FSW process on the aluminum alloys are researched systematically. The conventional FSW technologies, however, struggle to eliminate the microstructure difference between the bottom and top zone of the weld caused by the difference of the plastic deformation. Studies showed that the cracks were favored at the bottom surface of the stir zone, resulting in the reduction of mechanical properties of the joint [41]. The maturity of adopting FSW to joint magnesium alloys is still at an early stage in industrial application. Thus, it can be inferred that a higher mechanical property may be obtained through double-sided FSW processed carried out on both sides of the sample. In this study, double-sided FSW processes with different parameters were employed to finished the welding of the rolled AZ31 magnesium alloy. The effects of different welding parameters and manners on microstructure evolution and mechanical properties were studied. This work verified that the double-sided FSW improved the homogeneity of the welding joint microstructure and the mechanical properties. It contributes to promoting the application of the double-sided FSW on the weld assembly manufacture of the magnesium alloy sheet, such as the door frame, aircraft panel, and so on.

## 2. Materials and Methods

The rolled plate of AZ31 magnesium alloy having a thickness of 6 mm was chosen in this study, whose chemical composition is listed in Table 1. The rectangular workpieces with dimensions 100 mm × 100 mm were cut from the sheets. The contaminations adhering to the workpieces surface were removed by acetone cleaning before the welding process.

**Table 1.** Chemical composition of AZ31 rectangular workpieces (wt. %).

| Heading | Zn | Mg | Cu | Fe | Si | Mn | Ni | Other Total | Mg |
|---|---|---|---|---|---|---|---|---|---|
| Nominal | 0.6–1.4 | 2.5–3.5 | 0.05 | <0.05 | <0.1 | 0.1–0.2 | <0.05 | 0.3 | bal. |
| In this study | 1.2 | 3.3 | 0.04 | <0.05 | <0.1 | 0.14 | <0.05 | 0.27 | bal. |

As shown in Figure 1, the tool was manufactured using high-speed steel (HSS-H13) materials through the conventional lathe machine. The tool had a shoulder diameter of 15 mm, probe diameter of 8 mm, and length of 6 mm. The friction stir welding was conducted by the vertical milling machine (HT-JM8x23/2, Aerospace Engineering Equipment Co., Ltd., Suzhou, China). The FSW processes are shown in Table 2. Firstly, the workpiece is fixed on the worktable, and the tool is fixed in the spindle. Subsequently, the welding processes are carried out according to the parameters shown in Table 2.

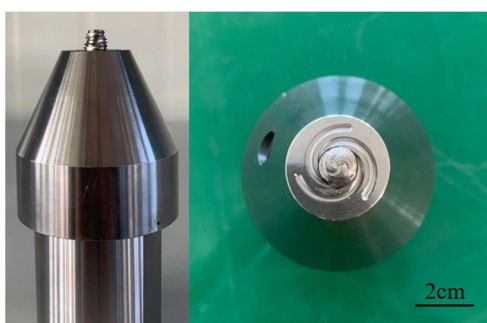

**Figure 1.** Photo of the friction stir tool.

**Table 2.** FSW process of AZ31 alloy.

| Sample | Rotation Speed/r/min | Welding Speed/mm/min | Welding Passes |
|---|---|---|---|
| BM | - | - | - |
| A | 500 | 60 | 1 |
| B | 1000 | 60 | 1 |
| C | 1500 | 60 | 1 |
| D | 500 | 60 | 2, double side |
| E | 500 | 60 | 2, Reciprocation |
| F | 1000 | 60 | 2, double side |
| G | 1000 | 60 | 2, Reciprocation |

After these FSW, observations under optical microscopy (OM, Leica DMLM, Buffalo Grove, IL, USA), scanning electron microscopy (SEM, JSM-6010, JOEL, Akishima, Japan), and electron backscattered diffraction (EBSD, Hikari XP, EDAX, San Diego, CA, USA) were carried out to analyze the microstructure of the joint center zones. The characterization of the microstructure was carried out at the cross-section of the joint. The samples for OM and SEM were ground with abrasive paper and then successively polished. The samples were etched using saturated picric acid reagent (4.2 g picric acid, 10 mL glacial acetic acid, 10 mL $H_2O$, and 70 mL of 95% ethanol) before observation. EBSD observations were performed on selected samples. The samples for EBSD were firstly mechanically polished through the similar process for optical observation. Then, the samples were electronically polished in the solution containing 10 mL perchloric acid and 90 mL ethanol for 20 s at −30 °C and 15 V. A Vickers hardness tester (FM-800, FUTURE-TECH, Tokyo, Japan) was employed to obtain the micro-hardness of different samples, operated with a load of 100 gf and holding time of 15 s. The samples prepared for micro-hardness were ground with abrasive paper and polished with 1.5 μm diamond compound. The tensile specimens were taken along the rolling direction and prepared according to Figure 2. The tensile tests were performed at room temperature using an MTS Landmark testing machine with a loading speed of

2 mm/min. There were three parallel testing samples of mechanical properties for every relevant FSW process.

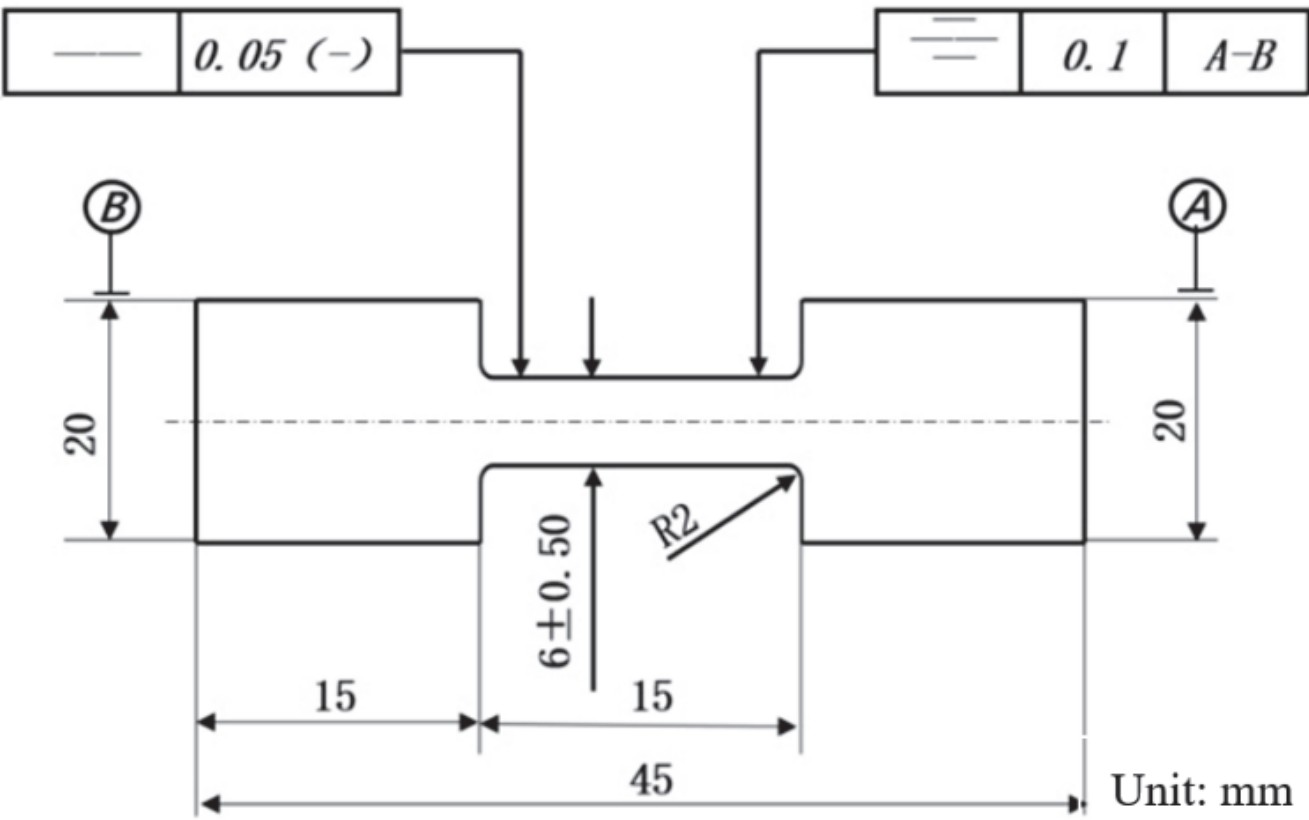

**Figure 2.** Schematic diagram with dimensions of the tensile test samples.

## 3. Results and Discussion

### 3.1. Effect of FSW Parameter on the Microstructure

3.1.1. The Effect of Rotate Speed on the Microstructure

Figure 3 shows the AZ31 alloy after an experienced FSW process with different parameters. It can be seen that the welding zones are well-formed with no defects in them.

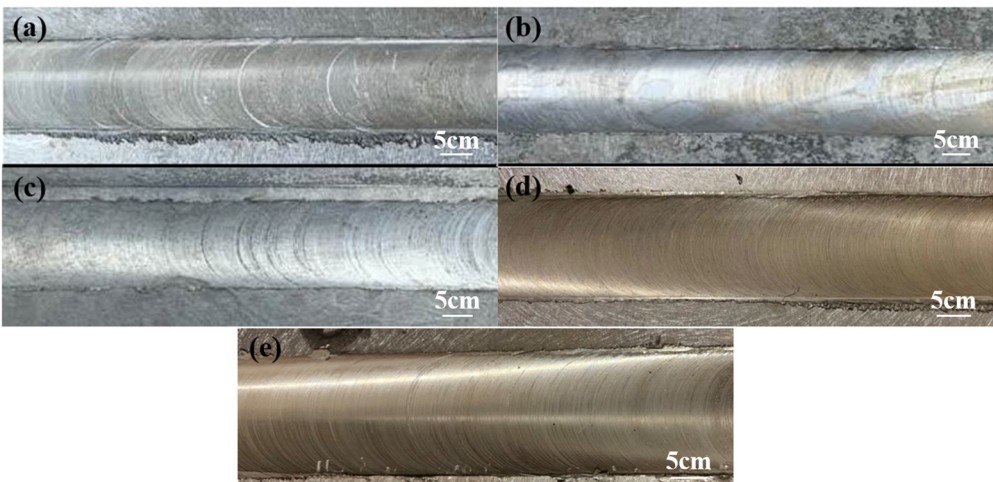

**Figure 3.** The photos of the joint after FSW with different rotation speeds: (**a**) 500 r/min; (**b**) 1000 r/min; (**c**) 1500 r/min; (**d**) 500 r/min double-sided; (**e**) 1000 r/min double-sided.

Figure 4 shows the microstructure of the center welding joint after experienced FSW

under different rotation speeds and base materials (BM) without the welding process. The base material presents the typical hot-rolling microstructure composed of minority fine equiaxed crystal and strip grain, as shown in Figure 4a. After experiencing the FSW, the microstructure at the center of the weld joint was mainly composed of the fine equiaxed grains (Figure 4b–d). According to Figure 4b, the weld joint with rotation speed of 500 r/min obtained a smaller grain size of 11.1 μm than BM sample (12.6 μm). When the rotation speed was increased to 1000 r/min, there was a slighter coarsening of the average grain size (11.6 μm). With the rotation increasing to 1500 r/min, the grain size grew to 12.1 μm.

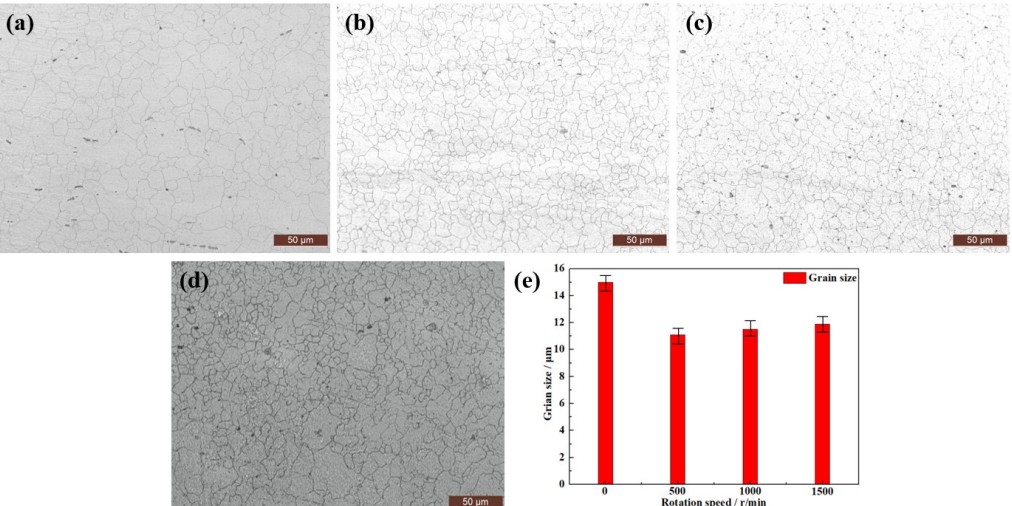

**Figure 4.** The optical microscope microstructure of the alloy after experiencing FSW with different rotation speeds: (**a**) original material; (**b**) 500 r/min; (**c**) 1000 r/min; (**d**) 1500 r/min; (**e**) distribution of the grain sizes.

As the typical low level fault energy materials (60–78 JM/m²), it is easiy to generate recrystallization for the magnesium alloy during the thermal deformation process [42]. In the FSW, the materials adjoining the weld tool underwent heating and severe deformation caused by the tool rotation, which induced the recrystallization and led to the appearance of fine equiaxed grains. Therefore, the microstructure was refined after experiencing the FSW. Heat input enlarged with the rotation speed increasing the weld tool, resulting in the deformation temperature increasing the joint materials. According to the work of Watanabe et al. [43], the grain size (d) of the magnesium alloy in FSW can be computed by Equations (1) and (2):

$$\ln Z = \dot{\varepsilon} exp(Q/RT) \tag{1}$$

$$\ln d = 9.0 - 0.27 \ln Z \tag{2}$$

where $Z$ is Zener Holfomon parameter, $Q$ is the thermal activation energy of the alloy, $R$ is the gas constant, and $T$ is the deformation temperature. From the above equations, it can be found that the grain size increase significantly with the increase of rotation speed of the weld tool.

### 3.1.2. The Effect of Welding Pass on the Microstructure

Figure 5 presents the microstructures of the samples after experiencing FSW with different parameters. It can be seen that the microstructure of the base material is composed of equiaxed grains and elongated shape grains. In the weld joints, after experiencing different FSW processes, there were three main zones namely stir zone (SZ), thermo-mechanical affected zone (TMAZ), and heat-affected zone (HAZ). The SZ is composed of fine and equiaxed grains formed through the complete dynamic recrystallization. Comparing with Figure 5a,c,e,g, the welding joint with a rotation speed of 500 r/min obtained a smaller



grain size than the sample welded with the rotation speed of 1000 r/min. Meanwhile, FSW carried out on double sides can obtain a better grain refinement than the FSW conducted by reciprocation. The minimum grain size is obtained in the sample with the FSW of D process. The SZ is completely affected by the heat input and mechanical deformation. The smaller grains in SZ showed that fierce straining took place in this zone. The heat input of the FSW (1000 r/min) is obviously larger than the FSW with the rotation speed of 500 r/min. Therefore, the grain growth of the weld joint with FSW (1000 r/min) obtained a bigger driving force, resulting in grain coarsening. From Figure 5b,d,f,i, the microstructure of HAZ showed relatively larger grains than SZ. The microstructure of the HAZ formed without any mechanical deformation was influenced only by the heat input of the tool. Consequently, the larger grains of the FSW (1000 r/min) were caused by the higher rotation speed than the sample of FSW (500 r/min). In the FSW, the deformation difference of the materials between the upper surface and the lower surface was ineluctable. In addition, the material flow in the FSW process can be divided into a horizontal flow and vertical flow. The double-sided FSW obviously can promote the vertical flow, enhancing the material strain at the joint. Hence, a more uniform and drastic deformation can obtain double-sided FSW more than the conventional multi-pass FSW conducted at the same surface, leading to a smaller grain size in double-sided FSW samples.

In order to further study the effect of the welding manner on the microstructure of the joint, distribution of the grain sizes and grain misorientation are statistically analyzed through the EBSD, as shown in Figure 6. It can be confirmed that the microstructure of the joint is affected significantly by the welding pass and the welding side (single or double side). A proportion of the fine grain was increasing with the application of a two-welding pass. The maximum ratio of fine grains was obtained in the sample, which experienced FSW carried out on double sides, respectively. As shown in Figure 6e, the ratio of grain misorientation (>10°) presents a similar variation tendency of the fine grains. Therefore, it can be concluded that grain refinement is mainly driven by the grain recrystallization mechanism of the magnesium alloy.

In the FSW, the deformation difference of the materials between the upper surface and the lower surface is ineluctable, which results in a wide distribution range of the grain size. The coarse grains could be broken firstly with the application of the next welding pass. Through controlling the time span between the welding times, the recrystallization temperature was suppressed effectively due to the good thermal conductivity of magnesium alloy. Therefore, the average grain size reduced with the increase of the welding pass. In addition, a better refinement efficiency was found in the welding process of the double-side due to the more uniform and serious deformation.

### 3.2. Effects of FSW Parameter on the Mechanical Properties

Figure 7 shows the microhardness distribution of the cross section of the samples along the width direction with different FSW. From Figure 7, it can be found that the microhardness presents a significant fluctuating character. The microhardness value of the cross section with D process is obviously higher than that of the sample with E process. Meanwhile, the microhardness curves of the cross section experienced the D process, which presents a lower fluctuation extent than the sample with the E process, which does agree with the microstructure character.

In order to study the effect of different parameters of FSW on mechanical properties, the tensile testing was performed after the samples experienced the welding, and the results are shown in Figure 8. Comparing with the BM (360 MPa), the mechanical properties of the samples present various degrees of reduction after experienced welding. Excluding the BM, meanwhile, the tendency of strength variation presents the same tend with the microstructure evolution. The welding joint experienced the FSW of the D process, which obtained the optimal strength (315 MPa).

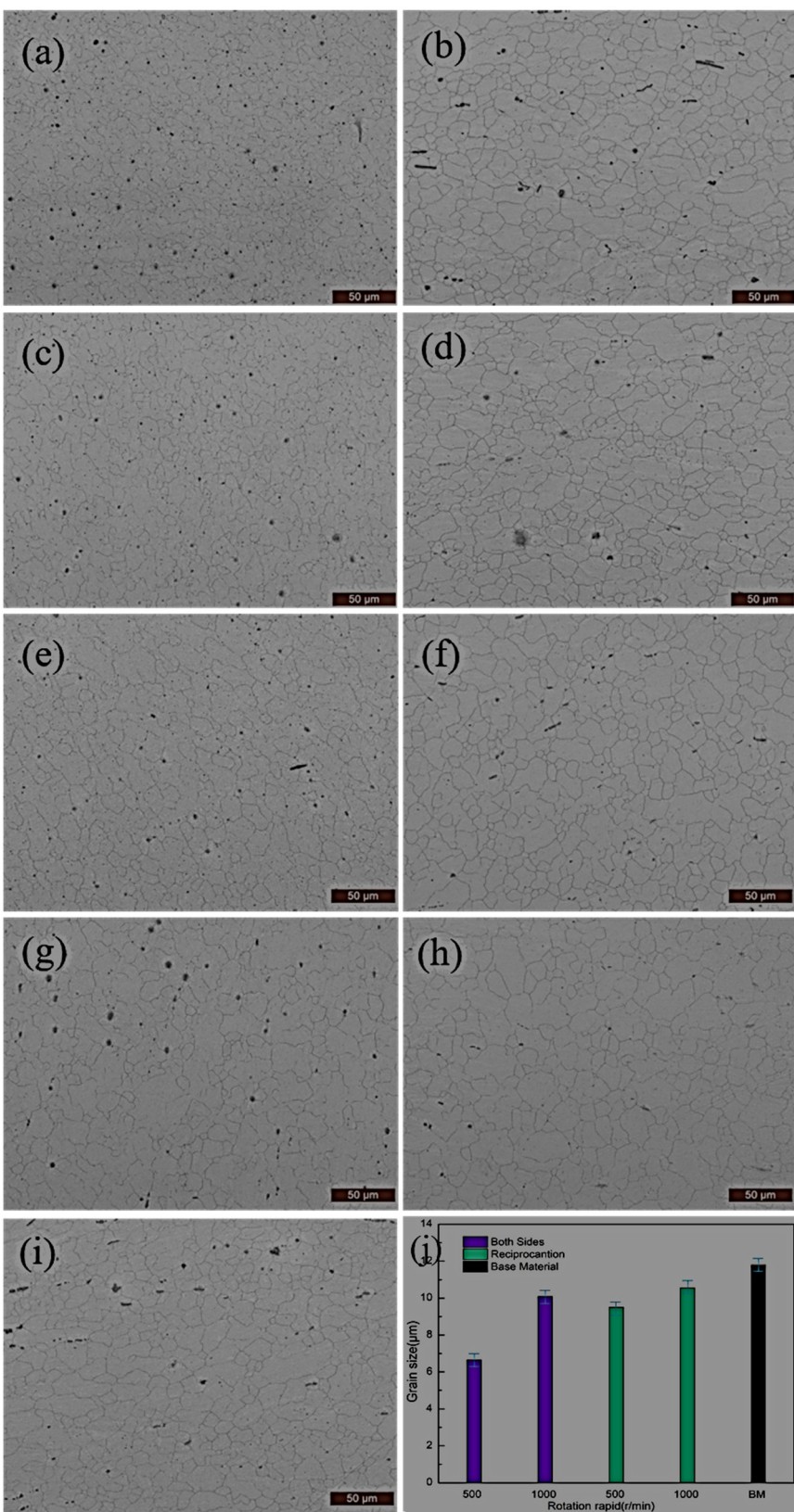

**Figure 5.** The microstructure of the weld joint after experiencing FSW with different welding parameters: Weld Nugget Zone: (**a**) 500 r/min, double-sided; (**c**) 500 r/min, reciprocation; (**e**) 1000 r/min, double-sided; (**g**) 1000 r/min, reciprocation. Heat Affected Zone: (**b**) 500 r/min, double-sided; (**d**) 500 r/min, reciprocation; (**f**) 1000 r/min, double-sided; (**h**) 1000 r/min, reciprocation; (**i**) base material; (**j**) distribution of the grain sizes.

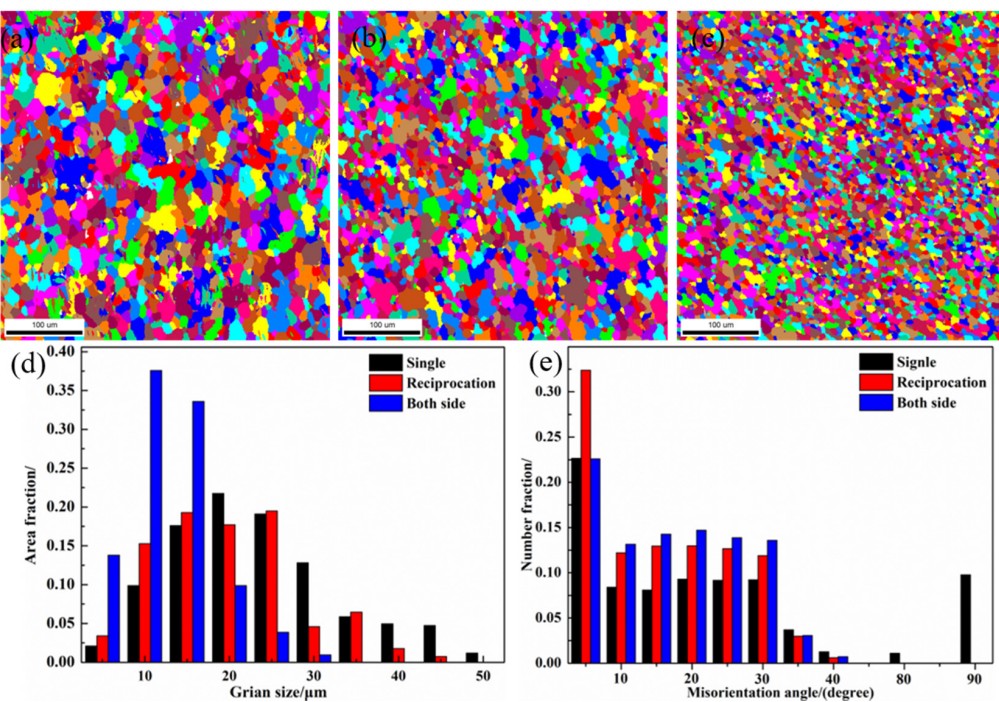

**Figure 6.** The EBSD results of the joint experienced different FSW with 500 r/min: (**a**) single; (**b**) reciprocation; (**c**) double-sided; (**d**) distribution of the grain sizes; (**e**) distribution of the grain misorientation.

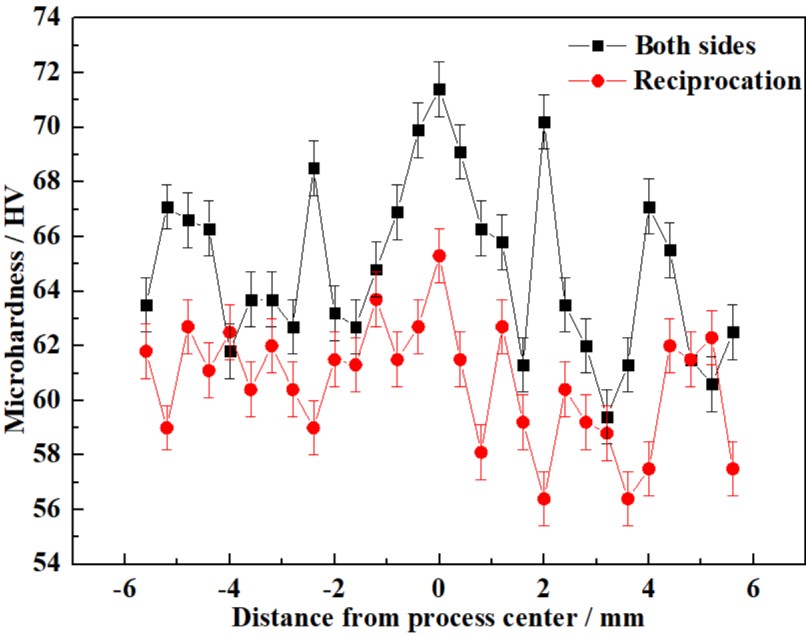

**Figure 7.** The microhardness distribution of the joint experienced the FSW process of D and E.

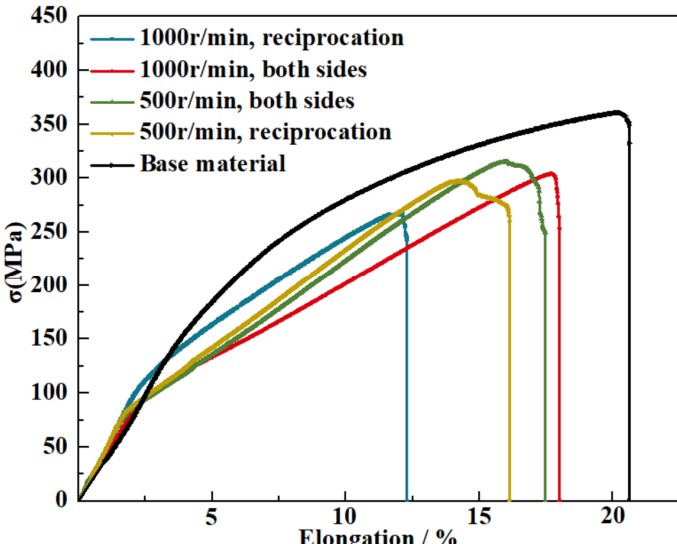

**Figure 8.** Tensile test result for the samples that experienced different FSWs.

According to the Hall–Petch equation, [34–47], the increment ($\Delta\sigma$) of the yield strength caused by grain refinement can be computed by Equation (3):

$$\sigma_y = \sigma_i + kd^{-2/1} \tag{3}$$

where $\sigma_y$ is the yield strength, $\sigma_i$ is the strength of pure aluminum, $d$ is the average grain size, and $k$ is a constant. The yield strength can increase significantly with the decline of grain size, resulting in the increase of micro-hardness. Comparing with the base materials, the mechanical properties were weakened by the generation of the heat-affected zone and enhanced by the refinement of the microstructure of the joint center. Therefore, the distribution of the microhardness presents the typical W type. In addition, a lower fluctuation range of microhardness and optimum mechanical properties in sample D were induced mainly by the uniform microstructure compared with other samples that experienced other welding processes.

However, when there is a sample with thick sections, there will be some difficult to eliminate the difference of the mechanical deformation between the bottom and top zone of the sample. Meanwhile, the difference of the mechanical deformation will be very slight when the sample thickness is lower than 3 mm in the FSW process. The double-sided FSW will be invalid.

## 4. Conclusions

(1) The grain refinement of the stirring zone can be induced by the FSW. In the single-pass FSW, the beneficial effect of FSW could be weakened by the increase of the rotation speed. The weld joint with a rotation speed of 500 r/min obtained the smallest grain size of 11.1 μm, which was smaller than the base materials (12.6 μm) and the joint of 1000 r/min (11.5 μm).

(2) The application of multi-pass welding in FSW induces a better refinement efficiency of the microstructure compared with the single pass. The microstructure refinement was carried out through the recrystallization mechanism.

(3) Compared with the reciprocation welding process, the multi-pass weld process conducted on double sides of the sample can induce a more uniform deformation, resulting in better grain refinement and a lower fluctuation of grain size. The maximum grain size (6.7 μm) was achieved on double-sided FSW of 500 r/min.

(4) The optimum mechanical properties (315 MPa) of the AZ31 alloy weld joints can be obtained through the multi-pass weld process conducted on double-sided FSW of 500 r/min.

**Author Contributions:** H.H. guided the investigation and experiments; S.C. analyzed the experiment data and mechanism, wrote the paper, and revised the paper; Y.Z. optimized the research plan and revised the paper. All authors have read and agreed to the published version of the manuscript.

**Funding:** This research received no external funding.

**Institutional Review Board Statement:** Not applicable.

**Informed Consent Statement:** Not applicable.

**Data Availability Statement:** Not applicable.

**Conflicts of Interest:** The authors declare no conflict of interest.

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
