# Peer review of "The Influence of the Mechanism of Double-Sided FSW on Microstructure and Mechanical Performance of AZ31 Alloy"

_metals, doi:10.3390/met11121982_

Round 1

Reviewer 1 Report

The following remarks can be made:

  • language needs attention, both in grammar and terminology. For example "...bounding process.." this is a joining process, "..rotate speed..." this is rotation speed, "...we can found.." grammar etc
  • literature review needs to expand and to demonstrate the need for the research presented in this manuscript
  • what is the make of the FSW machine
  • the title of figure 2 is wrong
  • figure 3d does not show that microstructure is affected by rotation speed changes
  • figure 3d and 4j are contradictory

Author Response

Dear Editor,

Thank you very much for considering our manuscript for review. We would like to appreciate the reviewers for their very constructive comments on our work and for providing valuable suggestions to further improve the quality of the article. We have studied the reviewers’ comments very carefully and have tried our best to revise our manuscript according to their comments. We have addressed all the reviewers’ comments and suggestions, as detailed below point by point. The related changes are highlighted in yellow in the revised manuscript.

We hope you will find that we’ve addressed all of the reviewers’ comments, and thus accept this article for publication in Materials.

Thank you for your time.

Yours faithfully,

Suna Cha, Hongliang Hou, Yanling Zhang

Reviewers' comments and our response to their comments (in blue):

  1. Comment: language needs attention, both in grammar and terminology. For example "...bounding process.." this is a joining process, "..rotate speed..." this is rotation speed, "...we can found.." grammar etc literature review needs to expand and to demonstrate the need for the research presented in this manuscript

Our response:

Thank you for your careful work. We have checked the manuscript carefully and revised the mistake of grammar and terminology in my new revision.

We noticed the problems in section of introduction, such as the unclear exposition of the research significance, less reference of the correlational studies. We gave a detailed modification in the revision edition. My modifications are as follows:

Please see the revision on page 1, line32: The limitation of joining magnesium alloys caused by the process of metallurgy and solidification in conventional welding technology can, therefore, be solved through the FSW process [9].

Page 2, line 45: FSP can refine grain structures and eliminate defects like porosity in the matrix of AZ31 magnesium alloy [14]. Barmouz found that FSP can refine the second particle’s size and enhance the dispersion distribution of the second particles for Cu/SiC composites [15].

Page 2, line 53: Bai et al. verified that joint strength could be improved significantly by the application of ultrasonic vibration during the FSW [20].

Page 2, line 58: Additionally, it has been reported that single-pass FSW requires a careful selection of welding parameters as improper welding parameters give rise to the formation of defects (such as cavity, tunnel and kissing bond defects) [27,28]. Compared to the single-pass FSW, the application of a second overlapping pass prolonged the DRX time and the DRX became sufficient, resulting in the further grain refinement [29]. Chen et al. found that reversing the welding direction of the second overlapping pass enhanced the vertical flow, increasing the FSW strain in the NZ [30].

Page 2, line 65: The conventional FSW technologies is, however, challenging to eliminate the microstructure difference between the bottom and top zone of the weld caused by the difference of the Plastic deformation. Studies showed that the cracks were favored at the bottom surface of the stir zone, resulting in the reduction of the mechanical properties of the joint [31]. The maturity of adopting FSW to joint magnesium alloys is still at an early stage in industrial application. Thus, it can be inferred that a higher mechanical property may be obtained through double-side FSW process carried out both side of the sample.

Reference:

[9] Z.Y. Ma; A.L. Pilchak, M.C. Juhas, J.C. Williams, Microstructural refinement and property enhancement of cast light alloys via friction stir processing. Scr. Mater. 2008, 58, 361–366.

[14] A. Gupta, P. Singh, P. Gulati, D.K. Shukla. Effect of Tool rotation speed and feed rate on the formation of tunnel defect in Friction Stir Processing of AZ31 Magnesium alloy. Mater. Today Proc. 2015, 2, 3463–3470.

[15] M. Barmouz, M.K.B Givi. Fabrication of in situ Cu/SiC composites using multi-pass friction stir processing: Evaluation of microstructural, porosity, mechanical and electrical behavior. Compos. Part A Appl. Sci. Manuf. 2011, 42, 1445–1453.

[20] Y.H. Bai, H. Su, C.S. Wu. Enhancement of the Al/Mg Dissimilar Friction Stir Welding Joint Strength with the Assistance of Ultrasonic Vibration. Metals. 2021,11,1113

[27] Y. Chen. H. Ding. J.Z. Li. Z.H. Cai. J.W. Zhao. W.J. Yang. Influence of multi-pass friction stir processing on the microstructure and mechanical properties of Al-5083 alloy. Mater. Sci. Eng. A 2016, 650, 281–289.

[28] N. Zhou, D.F. Song, W.J. Qi, X.Z. Li, J. Zou, M.M. Attallah. Influence of the kissing bond on the mechanical properties and fracture behaviour of AA5083-H112 friction stir welds. Mater. Sci. Eng. A 2018, 719, 12–20.

[29] X.C. Luo, L.M. Kang, H.L. Liu, J.Z. Li, Y.F. Liu, D.T. Zhang, D.L. Chen. Enhancing mechanical properties of AZ61 magnesium alloy via friction stir processing: Effect of processing parameters. Mater. Sci. Eng. A 2020, 797, 139945.

[30] Y. Chen, Z.H. Cai, H. Ding, F.H. Zhang. The Evolution of the Nugget Zone for Dissimilar AA6061/AA7075 Joints Fabricated via Multiple-Pass Friction Stir Welding. Metals, 2021, 11, 1506

[31] S. Mironov, T. Onum, Y.S. Sato, S. Yoneyama, H. Kokawa. Tensile behavior of friction-stir welded AZ31magnesium alloy. Mater. Sci. Eng. A 2017, 679, 272–281

  1. Comment: what is the make of the FSW machine?

Our response: We thank the reviewer for pointing out the missing of the make of the FSW machine. The make of the FSW machine has been presented in my revision. Please see the Page 2, line 84: HT-JM8x23/2, Aerospace Engineering Equipment Co., Ltd, Suzhou, China

  1. Comment: the title of figure 2 is wrong

Our response: Thank you for your careful work. We have checked the manuscript carefully and revised the mistake. Please see the Page 4, line 109: Fig. 2 Schematic diagram with dimensions of the tensile test samples.

  1. Comment: figure 3d does not show that microstructure is affected by rotation speed changes. figure 3d and 4j are contradictory.

Our response: We thank the reviewer for pointing out the mistake of the Fig.3 and Fig.4. The relevant microstructure of the two figures have been characterized and elaborated renewedly in my new revision.

Please see the Page 4, line 113: Fig. 3 shows the microstructure of the center welding joint after experienced FSW under different rotation speed and the base materials (BM) without weld process. The base material presents the typical hot-rolling microstructure composed of minority fine equiaxed crystal and strip grain, as shown in Fig. 3a. After experienced the FSW, the microstructure at the center of the weld joint was mainly composed of the fine equiaxed grains (Fig. 3b, 3c and 3d). According to the Fig. 3b, the weld joint with rotation speed of 500 r/min obtained a smaller grain size of 11.1 μm than BM sample (12.6μm). When the rotation speed is increased to 1000 r/min, there is slighter coarsening of the average grain size (11.6 μm). With the rotation increasing to 1500r/min, the grain size will grow to 12.1 μm.

Fig. 3 The optical microscope microstructure of the alloy after experienced FSW with different rotate speed: (a) original material; (b) 500r/min; (c) 1000r/min; (d) 1500r/min; (e) distribution of the grain sizes.

Page 5, line 144: It can be seen that the microstructure of the base material is composed of equiaxed grains and elongated shape grains.

Page 5, line 152: The minimum grain size is obtained in the sample with the FSW of D process.

Page 5, line 163: In addition, the material flow in the FSW process can be divided into a horizontal flow and vertical flow. The double-side FSW can promote obviously the vertical flow, enhancing the material strain at the joint. Hence, a more uniform and drastic deformation can be obtained double-side FSW than the conventional multi-pass FSW conducted at the same surface, leading to the smaller grain size in double-size FSW samples.

Reviewer 2 Report

The paper could be accepted as-is.

Author Response

Dear Editor,

Thank you very much for considering our manuscript for review. We would like to appreciate the reviewers for their very constructive comments on our work and for providing valuable suggestions to further improve the quality of the article. We have studied the reviewers’ comments very carefully and have tried our best to revise our manuscript according to their comments. We have addressed all the reviewers’ comments and suggestions, as detailed below point by point. The related changes are highlighted in yellow in the revised manuscript.

We hope you will find that we’ve addressed all of the reviewers’ comments, and thus accept this article for publication in Materials.

Thank you for your time.

Yours faithfully,

Suna Cha, Hongliang Hou, Yanling Zhang

Reviewers' comments and our response to their comments (in blue):

Comments and Suggestions:

The paper could be accepted as-is.

Our response:

Thank you very much for your comment, and affirmation for my work. I will improve my work in the future, and make it more innovative. I really appreciate your careful work.

Reviewer 3 Report

Manuscript is quite interesting to read. But in my point of view, I found some queries that I can mention here in the following points.

  1. Research gap/purpose of the research shall be explained well.
  2. The control samples considered was base material and grin size after weld was investigated but not reflected in the conclusion
  3. This investigation verified basic truth of FSW and Welding Technology
  4. Justify that As welded sample thickness 6 mm considered why double side weld and multi-pass weld required? (Any Targeted application? / in efficient of FSW?)
  5. As FSW merit is better grain size after weld what is the use of verifying the same?
  6. AS No welded Samples photos presented; how can we justify philosophy discussed?
  7. Is there any significant variation on grain structure by welding both side at 1000 rpm stirring speed?
  8. Only one process parameter and 500 and 1000 only considered is it sufficient?
  9. Limitations of the study shall be discussed.
  10. Is double side weld improving targeted application requirement like tensile strength? if so is it doubles?
  11. As many related investigation published in this scope by the journal of “metals”. Those articles shall be referred and cited suitably.
  12. Kindly include this reference

    M. Kavitha, V. M. Manickavasagam, T.Sathish, Bhiksha Gugulothu, A. Sathish Kumar, Sivakumar Karthikeyan and Ram Subbiah ‘Parameters Optimization of Dissimilar Friction Stir Welding for AA7079 and AA8050 through RSM, Advances in Materials Science and Engineering, Volume 2021, Article ID 9723699, https://doi.org/10.1155/2021/9723699.

Author Response

November 25, 2021

Metals

Dear Editor,

Thank you very much for considering our manuscript for review. We would like to appreciate the reviewers for their very constructive comments on our work and for providing valuable suggestions to further improve the quality of the article. We have studied the reviewers’ comments very carefully and have tried our best to revise our manuscript according to their comments. We have addressed all the reviewers’ comments and suggestions, as detailed below point by point. The related changes are highlighted in yellow in the revised manuscript.

We hope you will find that we’ve addressed all of the reviewers’ comments, and thus accept this article for publication in Materials.

Thank you for your time.

Yours faithfully,

Suna Cha, Hongliang Hou, Yanling Zhang

Reviewers' comments and our response to their comments (in blue):

Comments and Suggestions for Authors

Manuscript is quite interesting to read. But in my point of view, I found some queries that I can mention here in the following points.

  1. Comment: Research gap/purpose of the research shall be explained well.

Our response: Thank you very much for your comments. We noticed the unclear exposition of the research purpose. We gave a detailed modification in the revision edition. My modifications are as follows:

Please see the revision on page 1, line32: The limitation of joining magnesium alloys caused by the process of metallurgy and solidification in conventional welding technology can, therefore, be solved through the FSW process [9].

Please see the revision on page 2, line45: FSP can refine grain structures and eliminate defects like porosity in the matrix of AZ31 magnesium alloy [14].

Please see the revision on page 2, line65: The conventional FSW technologies is, however, challenging to eliminate the microstructure difference between the bottom and top zone of the weld caused by the difference of the Plastic deformation. Studies showed that the cracks were favored at the bottom surface of the stir zone, resulting in the reduction of the mechanical properties of the joint [32]. The maturity of adopting FSW to joint magnesium alloys is still at an early stage in industrial application. Thus, it can be inferred that a higher mechanical property may be obtained through double-side FSW process carried out both side of the sample.

  1. Comment: The control samples considered was base material and grin size after weld was investigated but not reflected in the conclusion.

Our response: We thank the reviewer for pointing out the missing of the grain size in conclusion. The conclusions have been revised carefully in the new revision. Please see the Page 8, line 224 to 234:

(1) The grain refinement of stirring zone can be induced by the FSW. In the single pass FSW, the beneficial effect of FSW could be weakened by the increase of the rotation speed. The weld joint with rotation speed of 500 r/min obtained the smallest grain size of 11.1 μm which was smaller than the base materials (12.6 μm) and the joint of 1000r/min (11.5 μm).

(3) Compared with the reciprocation welding process, the multi-pass weld process conducted at double side of the sample can induce a more uniform deformation, resulting in better grain refinement and a lower fluctuation of grain size. The maximum grain size (6.7 μm) was achieved at double-side FSW of 500r/min.

  1. Comment: This investigation verified basic truth of FSW and Welding Technology.

Our response: We thank you for providing the query. The mechanical properties are lower than the original material, leading to limitation of the widely application of wrought magnesium alloy. After experienced several decades development, the theories and parameters obtained systematic research. However, those studies are mainly performed with the variation of welding passes and travel direction. Consequently, we study the effect of double-side FSW on the mechanical properties and microstructure of the joint. To our pleasure, the microstructure and strength of the joint was improvement.

  1. Comment: Justify that As welded sample thickness 6 mm considered why double side weld and multi-pass weld required? (Any Targeted application? / in efficient of FSW?)

Our response: Thank you very much for your careful work. The FSW technology can solve the limitation of the conventional welding technology caused by the process of metallurgy and solidification. The conventional FSW technologies is, however, challenging to eliminate the microstructure difference between the bottom and top zone of the weld caused by the difference of the Plastic deformation, which deteriorate the mechanical properties of the joint. This limitation could be eliminated obviously by when the FSW was performed on the both side of the samples. Compared to the single-pass FSW, in addition, the application of a second overlapping pass prolonged the DRX time and the DRX became sufficient, resulting in the further grain refinement.

  1. Comment: As FSW merit is better grain size after weld what is the use of verifying the same?

Our response: We thank you for providing the query. In the conventional FSW technologies, the microstructure difference between the bottom and top zone of the joint is disadvantage of the mechanical properties. Meanwhile, microstructure difference can promote the origin and growth of crack. However, this problem can be solved by the double-side FSW technology through improve the microstructure difference. 

  1. Comment: AS No welded Samples photos presented; how can we justify philosophy discussed?

Our response: We thank the reviewer for pointing out the missing of the welded samples photos. The relevant photos and result and explanation had been add in the revision.

Please see Page 4, line 113: Fig. 3 shows the AZ31 alloy after experienced FSW process with different parameters. It can be seen that the welding zones are well-formed with no defects in it.

Fig. 3 The photos of the joint after FSW with different rotation speed:

(a) 500r/min; (b) 1000r/min; (c) 1500r/min; (d) 500r/min double-side; (e) 1000r/min double-side.

  1. Comment: Is there any significant variation on grain structure by welding both side at 1000 rpm stirring speed?

Our response: Thank you very much for your careful work. Comparing with the single pass FSW, a smaller grain size can be obtained in the multi-pass FSW. However, there is only a slight difference between two passes FSW performed at the same side and the double-side FSW. The high heat input under rotation speed of 1000r/min reduce the refinement effect of the double-side FSW technology.

  1. Comment: Only one process parameter and 500 and 1000 only considered is it sufficient?

Our response: Thank you very much for your comments. When the FSW was conducted with the rotation of 350r/min, the frictional heat is not sufficient to promote joint material flow, resulting in the failure of weld with several macro-defects like porosity, hot cracks, voids, and tunnels. Hence, the result of 350r/min was not shown in this article. In order to give a sufficient data support, the result of the FSW with the rotation speed of 1500r/min was shown in the section 3 (results and discussion).

.

  1. Comment: Limitations of the study shall be discussed.

Our response: Thank you very much for your advice to add a Limitations discussion of this study. When the sample with thick sections, there will be some difficult to eliminate the difference of the mechanical deformation between the bottom and top zone of the sample. Meanwhile, the difference of the mechanical deformation will be very slight when the sample thickness is lower than 3mm in FSW process. The double-side FSW will be invalid.

Please see the page 8, line 223.

  1. Comment: Is double side weld improving targeted application requirement like tensile strength? if so is it doubles?

Our response: Thank you very much for your comments. In the FSW process, the material flow can be divided into a horizontal flow and vertical flow. The double-side FSW can promote obviously the vertical flow, enhancing the material strain at the joint. Hence, a more uniform and drastic deformation can be obtained double-side FSW than the conventional multi-pass FSW, leading to the smaller grain size and better strength. In the double-side FSW, however, some degree improvement can be obtained, not doubles.

  1. Comment: As many related investigation published in this scope by the journal of “metals”. Those articles shall be referred and cited suitably.

Our response: We thank you for providing the advice to add some reference by the journal of Metals. Some references has been cited in this revision. Please see Page 9 and Page 10:

[20] Y.H. Bai, H. Su, C.S. Wu. Enhancement of the Al/Mg Dissimilar Friction Stir Welding Joint Strength with the Assistance of Ultrasonic Vibration. Metals. 2021,11,1113.

[31] Y. Chen, Z.H. Cai, H. Ding, F.H. Zhang. The Evolution of the Nugget Zone for Dissimilar AA6061/AA7075 Joints Fabricated via Multiple-Pass Friction Stir Welding. Metals, 2021, 11, 1506.

  1. Comment: Kindly include this reference. M. Kavitha, V. M. Manickavasagam, T.Sathish, Bhiksha Gugulothu, A. Sathish Kumar, Sivakumar Karthikeyan and Ram Subbiah ‘Parameters Optimization of Dissimilar Friction Stir Welding for AA7079 and AA8050 through RSM, Advances in Materials Science and Engineering, Volume 2021, Article ID 9723699, https://doi.org/10.1155/2021/9723699.

Our response: Thank you for providing the advice to add this reference. This reference has been read carefully and cited in the section of introduction. Please see Page 2, line 58: Kavitha et al. studied the effect of FSW parameters on the joint strength of AA7079 and AA8050 through a statistical technique of RSM, and found the preferred process parameters [29].

[29] M. Kavitha, V. M. Manickavasagam, T.Sathish, B. Gugulothu, A. S. Kumar, S. Karthikeyan, R. Subbiah. Parameters Optimization of Dissimilar Friction Stir Welding for AA7079 and AA8050 through RSM, Adv. Mater. Sci. Eng.. Vol. 2021, Article ID 9723699, https://doi.org/10.1155/2021/9723699

Round 2

Reviewer 1 Report

The literature review still lacks the references to AZ31.

Author Response

  1. Kavitha, V. M. Manickavasagam, T.Sathish, Bhiksha Gugulothu, A. Sathish Kumar, Sivakumar Karthikeyan and Ram Subbiah ‘Parameters Optimization of Dissimilar Friction Stir Welding for AA7079 and AA8050 through RSM, Advances in Materials Science and Engineering, Volume 2021, Article ID 9723699, https://doi.org/10.1155/2021/9723699.
  2. Jayaprakash, S. Siva Chandran, T. Sathish, Bhiksha Gugulothu, R. Ramesh,  M. Sudhakar and Ram Subbiah ‘Effect of Tool Profile Influence in Dissimilar Friction Stir Welding of Aluminium Alloys (AA5083 and AA7068), Advances in Materials Science and Engineering, Volume 2021, Article ID 7387296, https://doi.org/10.1155/2021/7387296.

Reviewer 3 Report

Manuscript is quite interesting to read and paper having adequate information.

Influence the Mechanism of Double-side FSW is more noval.

Kindly Proof read your manuscript.

Include conclusion part at the end of abstract.

Error bar is missing.

Kindly specify the application for this study.

Kindly include these reference.

M. Kavitha, V. M. Manickavasagam, T.Sathish, Bhiksha Gugulothu, A. Sathish Kumar, Sivakumar Karthikeyan and Ram Subbiah ‘Parameters Optimization of Dissimilar Friction Stir Welding for AA7079 and AA8050 through RSM, Advances in Materials Science and Engineering, Volume 2021, Article ID 9723699, https://doi.org/10.1155/2021/9723699.

S. Jayaprakash, S. Siva Chandran, T. Sathish, Bhiksha Gugulothu, R. Ramesh,  M. Sudhakar and Ram Subbiah ‘Effect of Tool Profile Influence in Dissimilar Friction Stir Welding of Aluminium Alloys (AA5083 and AA7068), Advances in Materials Science and Engineering, Volume 2021, Article ID 7387296, https://doi.org/10.1155/2021/7387296.

Author Response

December 01, 2021

Metals

Dear Editor,

Thank you very much for considering our manuscript for review. We would like to appreciate the reviewers for their very constructive comments on our work and for providing valuable suggestions to further improve the quality of the article. We have studied the reviewers’ comments very carefully and have tried our best to revise our manuscript according to their comments. We have addressed all the reviewers’ comments and suggestions, as detailed below point by point. The related changes are highlighted in yellow in the revised manuscript.

We hope you will find that we’ve addressed all of the reviewers’ comments, and thus accept this article for publication in Metals.

Thank you for your time.

Yours faithfully,

Suna Cha, Hongliang Hou, Yanling Zhang

Reviewers' comments and our response to their comments (in blue):

  1. Comment: Influence the Mechanism of Double-side FSW is more noval.

Our response: Thank you very much for your comments. We

  1. Comment: Kindly Proof read your manuscript. Include conclusion part at the end of abstract.

Our response: Thank you very much for your comments. We did proof read the manuscript and gave a detail revision. Please see the new revision.

  1. Comment: Error bar is missing.

Our response: We thank you for pointing the missing of the error bar in the figures. In the new revision, Error bar has been add in the relevant figures. Please see the new revision:

Fig. 4 The optical microscope microstructure of the alloy after experienced FSW with different rotate speed: (a) original material; (b) 500r/min; (c) 1000r/min; (d) 1500r/min; (e) distribution of the grain sizes.

Fig. 5 The microstructure of the weld joint after experienced FSW with different welding parameters: Weld Nugget Zone: (a) 500r/min, double side; (c) 500r/min, reciprocation; (e) 1000r/min, double side; (g) 1000r/min, reciprocation; Heat Affected Zone: (b) 500r/min, double side; (d) 500r/min, reciprocation; (f) 1000r/min, double side; (h) 1000r/min, reciprocation; (i) base material; (j)distribution of the grain sizes.

Fig. 7 The microhardness distribution of the joint experienced the FSW process of D and E

  1. Comment: Kindly specify the application for this study.

Our response: Thank you very much for your careful work. The application of this investigation has been explained in the new revision. Please see the revision at page 2, line 76:

This work verified that the double-side FSW improved the homogeneity of the welding joint microstructure and the mechanical properties. It contributes to promote the application of the double-side FSW on the weld assembly manufacture of the magnesium alloy sheet, such as the door frame, aircraft panel and so on.

  1. Comment: Kindly include these references.
  2. Kavitha, V. M. Manickavasagam, T.Sathish, Bhiksha Gugulothu, A. Sathish Kumar, Sivakumar Karthikeyan and Ram Subbiah ‘Parameters Optimization of Dissimilar Friction Stir Welding for AA7079 and AA8050 through RSM, Advances in Materials Science and Engineering, Volume 2021, Article ID 9723699, https://doi.org/10.1155/2021/9723699.
  3. Jayaprakash, S. Siva Chandran, T. Sathish, Bhiksha Gugulothu, R. Ramesh,  M. Sudhakar and Ram Subbiah ‘Effect of Tool Profile Influence in Dissimilar Friction Stir Welding of Aluminium Alloys (AA5083 and AA7068), Advances in Materials Science and Engineering, Volume 2021, Article ID 7387296, https://doi.org/10.1155/2021/7387296.

Our response: Our response: Thank you for providing the advice to add those reference. Those reference has been read carefully and cited in the section of introduction. Please see Page 11, line 309:

  1. Jayaprakash, S. Siva Chandran, T. Sathish, Bhiksha Gugulothu, R. Ramesh, M. Sudhakar and Ram Subbiah ‘Effect of Tool Profile Influence in Dissimilar Friction Stir Welding of Aluminium Alloys (AA5083 and AA7068), Advances in Materials Science and Engineering, Volume 2021, Article ID 7387296, https://doi.org/10.1155/2021/7387296. [22]
  2. Kavitha, V. M. Manickavasagam, T.Sathish, Bhiksha Gugulothu, A. Sathish Kumar, Sivakumar Karthikeyan and Ram Subbiah ‘Parameters Optimization of Dissimilar Friction Stir Welding for AA7079 and AA8050 through RSM, Advances in Materials Science and Engineering, Volume 2021, Article ID 9723699, https://doi.org/10.1155/2021/9723699. [31]
